# Neural Attribution for Semantic Bug-Localization in Student Programs

**Rahul Gupta**[1]     **Aditya Kanade**[1,2]     **Shirish Shevade**[1]

[1]Department of Computer Science and Automation,
Indian Institute of Science, Bangalore, KA 560012, India
[2]Google Brain, CA, USA
{rahulg, kanade, shirish}@iisc.ac.in

## Abstract

Providing feedback is an integral part of teaching. Most open online courses on programming make use of automated grading systems to support programming assignments and give real-time feedback. These systems usually rely on test results to quantify the programs' functional correctness. They return failing tests to the students as feedback. However, students may find it difficult to debug their programs if they receive no hints about where the bug is and how to fix it. In this work, we present NeuralBugLocator, a deep learning based technique, that can localize the bugs in a faulty program with respect to a failing test, without even running the program. At the heart of our technique is a novel tree convolutional neural network which is trained to predict whether a program passes or fails a given test. To localize the bugs, we analyze the trained network using a state-of-the-art neural prediction attribution technique and see which lines of the programs make it predict the test outcomes. Our experiments show that NeuralBugLocator is generally more accurate than two state-of-the-art program-spectrum based and one syntactic difference based bug-localization baselines.

## 1   Introduction

Automated grading systems for student programs both check the functional correctness of assignment submissions and provide real-time feedback to students. The feedback helps students learn from their mistakes, allowing them to revise and resubmit their work. In the current practice, automated grading systems rely on running the submissions against a test suite. The failing tests are returned to the students as feedback. However, students may find it difficult to debug their programs if they receive no hints about where the bug is and how to fix it. Although instructors may inspect the code and manually provide such hints in a traditional classroom setting, doing this in an online course with a large number of students is often infeasible. Therefore, our aim in this work is to develop an automated technique for generating feedback about the error locations corresponding to the failing tests. Such a technique benefits both instructors and students by allowing instructors to automatically generate hints for students without giving away the complete solution.

Towards this, we propose a deep learning based semantic bug-localization technique. While running a program against a test suite can detect the presence of bugs in the program, locating these bugs requires careful analysis of the program behavior. Our proposed technique can localize the bugs in a buggy program with respect to a failing test, without even running the program. It works in two phases. In the first phase, we train a novel tree convolutional neural network to predict whether or not a program passes a given test. The input to this network is a pair of a program and a test ID. In the second phase, we query a state-of-the-art neural prediction attribution technique [25] to find out which lines of a buggy program make the network predict the failure to localize the bugs. We call our

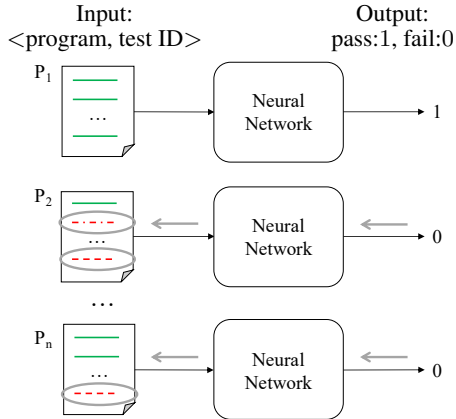

Input: <program, test ID>

Output: pass:1, fail:0

```
1.  #include <stdio.h>
2.  int main(){
3.    char c;
4.    scanf("%c", &c);
5.    if('A'<=c && c<='Z'){
6.      printf("%c", c+('a'-'A'));}
7.    else if ('a'<=c && c<='z'){
8.      printf("%c", c-('a'-'A'));}
9.    else if(0<=c && c<=9){
10.     printf("%d", 9-'c');}
11.   else{
12.     printf("%c", c);}
13.   return 0;}
```

Figure 1: Overview of NBL. The buggy lines in the input programs are represented by dashed lines. We omit test IDs from the input for brevity. The forward black arrows show the neural network prediction for each input. The thick gray arrows and ovals show the prediction attribution back to the buggy input programs leading to bug-localization.

Figure 2: Example to illustrate the NBL approach. The program shown has bugs at lines 9 and 10 (shown in bold). The top-5 suspicious lines returned by NBL for this program are marked using a heat-map where darker color indicates higher suspiciousness score.

technique *NeuralBugLocator* or *NBL* in short. Figure 1 shows the overview of NBL. Figure 2 shows NBL in action on a buggy student submission with respect to a failed test. For a given character $c$, the programming task for this submission requires as output the character obtained by reversing its case if $c$ is a letter of the alphabet, or $9 - c$ if it is a digit, otherwise c itself. The illustrated submission mishandles the second case in lines 9 and 10 and prints the same digit as the input.

Prediction attribution techniques are employed for attributing the prediction of a deep network to its input features. For example, for a multi-class image recognition network, a prediction attribution technique can identify the pixels associated with the given class in the given image, and thus can be used for object-localization in spite of being trained on image labels only. Our work introduces prediction attribution for semantic bug-localization in programs.

Bug-localization is an active field of research in software engineering [29]. Spectrum-based bug localization approach [14, 2] instruments the programs to get program traces corresponding to both the failing and the passing tests. In order to locate the bugs in a program, it compares the program statements that are executed in failing test runs against those that are executed in passing test runs. While spectrum-based bug-localization exploits correlations between executions of the same program on multiple tests, our technique exploits similarities and differences between the code of multiple programs with respect to the same test. In this way, the former is a dynamic program analysis approach, whereas the latter is a static program analysis approach.

The existing static approaches for bug-localization in student programs compare a buggy program with a reference implementation [15, 16]. However, the buggy program and the reference implementation can use different variable names, constant values, and data and control structures, making it extremely difficult to distinguish bug inducing differences from the benign ones. Doing this requires the use of sophisticated program analysis techniques along with heuristics, which may not work for a different programming language. In contrast, NBL does not require any heuristics and therefore, is programming language agnostic.

Use of machine learning in software engineering research is not new. Several recent works proposed deep learning based techniques for automated syntactic error repair in student programs [11, 6, 3, 13]. Bugram [28] is a language model based bug-detection technique. Pu et al. [21] propose a deep learning based technique for both syntactic and semantic error repair in small student programs. Their technique uses a brute-force, enumerative search for detecting and localizing bugs. Another recent work [26] proposed a multi-headed LSTM pointer network for joint localization and repair of variable-misuse bugs. In contrast, ours is a semantic bug-localization technique, which learns to find

the location of the buggy statements in a program. Unlike these approaches, our technique neither requires explicit bug-localization information for training nor does it perform a brute-force search. Instead, it trains a neural network to predict whether or not a program passes a test and analyses gradients of the trained network for bug-localization. Moreover, our technique is more general and works for all kinds of semantic bugs. To the best of our knowledge, we are the first to propose a general deep learning technique for semantic bug-localization in programs w.r.t. failing tests.

We train and evaluate NBL on C programs written by students for 29 different programming tasks in an introductory programming course. The dataset comes with 231 instructor written tests for these tasks. Thus, programs for each task are tested against about 8 tests on an average. We compare NBL with three baselines which include two state-of-the-art, program-spectrum based techniques [14, 2] and one syntactic difference based technique. Our experiments demonstrate that NBL is more accurate than them in most cases. The main contributions of this work are as follows:

1. It proposes a novel encoding of program ASTs and a tree convolutional neural network that allow efficient batch training for arbitrarily shaped trees.

2. It presents the first deep learning based general technique for semantic bug-localization in programs. It also introduces prediction attribution in the context of programs.

3. The proposed technique is evaluated on thousands of buggy C programs with encouraging results. It successfully localized a wide variety of semantic bugs, including wrong conditionals, assignments, output formatting and memory allocation, among others.

4. We provide both the dataset and the implementation of NBL online at `https://bitbucket.org/iiscseal/nbl/`.

## 2 Background: prediction attribution

Prediction attribution techniques attribute the prediction of a deep network to its input features. For our task of bug-localization, we use a state-of-the-art prediction attribution technique called integrated gradients [25]. This technique has been shown to be effective in domains as diverse as object recognition, medical imaging, question classification, and neural machine translation among others. In Section 3.2, we explain how we leverage integrated gradients for bug-localization in programs. Here we describe this technique briefly. For more details, we refer our readers to the work of Sundararajan et al. [25].

When assigning credit for a prediction to a certain feature in the input, the absence of the feature is required as a baseline for comparing outcomes. This absence is modeled as a single baseline input on which the prediction of the neural network is "neutral" i.e., conveys a complete absence of signal. For example, in object recognition networks, the black image can be considered as a neutral baseline. Integrated gradients technique distributes the difference between the two outputs (corresponding to the input of interest and the baseline) to the individual input features.

More formally, for a deep network representing a function $F : \mathbb{R}^n \to [0, 1]$, input $x \in \mathbb{R}^n$, and baseline $x' \in \mathbb{R}^n$; integrated gradients are defined as the path integral of the gradients along the straight-line path from the baseline $x'$ to the input $x$. For $x$ and $x'$, the integrated gradient (IG) along the $i^{th}$ dimension is defined as follows:

$$\mathsf{IG}_i(x) = (x_i - x_i') \times \int_{\alpha=0}^{1} \frac{\partial F(x' + \alpha(x - x'))}{\partial x_i} \, d\alpha$$

If $F : \mathbb{R}^n \to \mathbb{R}$ is differentiable almost everywhere, then it can be shown that:

$$\sum_{i=1}^{n} \mathsf{IG}_i(x) = F(x) - F(x')$$

If the baseline $x'$ is chosen in a way such that the prediction at the baseline is near zero ($F(x') \approx 0$), then resulting attributions have an interpretation that ignores the baseline and amounts to distributing the output to the individual input features. The integrated gradients can be efficiently approximated via summing the gradients at points occurring at sufficiently small intervals along the straight-line path from the baseline $x'$ to the input $x$, with $m$ being the number of steps in the Riemman approximation of the integral of integrated gradients [25].

$$\mathsf{IG}_i^{\text{approx}}(x) = (x_i - x_i') \times \sum_{k=1}^{m} \frac{\partial F(x' + \frac{k}{m}(x - x')))}{\partial x_i} \times \frac{1}{m}$$

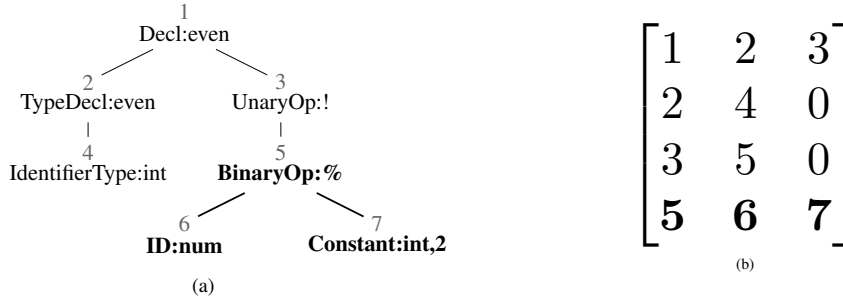

(a)

(b)

Figure 3: 3a: AST of the code snippet: `int even = !(num%2)`. For each node, its visiting order is also shown in the breadth-first traversal of the AST. 3b: 2D matrix representation of the AST shown in Figure 3a. The matrix shows node positions instead of the nodes themselves to avoid clutter. For example, the last row corresponds to the highlighted subtree from Figure 3a.

## 3 Technical details

We divide our bug-localization approach into two phases. In the first phase, we train a neural network to predict whether or not a program passes the test corresponding to a given test ID. This is essentially a classification problem with two inputs: program text and a test ID, where we have multiple passing and failing programs (which map to different class labels) for each test ID. Though different programs are used at the time of evaluation, they share test IDs with the training examples. Alternatively, this can be thought of as task-conditional multi-task learning with test IDs identifying the tasks. In the second phase, we perform bug-localization by identifying patterns that help the neural network in correct classification. Note that the neural network is only given the test ID along with the program as input. It is not provided with the actual inputs and the corresponding outputs of the tests as it does not know how to execute programs. The learning is based only on the presence or absence of syntactic patterns in the programs.

### 3.1 Phase 1: tree convolutional neural network for test failure prediction

Use of machine learning in software engineering research is not new [4]. Many existing works which use machine learning algorithms on programs use recurrent neural networks (RNNs) [11, 26] and convolutional neural networks (CNNs) [18]. Our initial experiments with multiple variants of both RNNs and CNNs suggested the latter to be better suited for our task. CNNs are designed to capture spatial neighborhood information in data, and are generally used with inputs having a grid-like structure, such as images [9]. On their own, they may fail to capture the hierarchical structures present in programs. To address this, Mou et al. [18] proposed tree based CNNs. However, the design of their custom filter is difficult to implement and train as it does not allow batch computation over variable-sized programs and trees. Therefore, we propose a novel tree convolutional network which uses specialized program encoding and convolution filters to capture the tree structural information present in programs, allowing us to not only batch variable-sized programs but also leverage the well-optimized CNN implementations provided by popular deep learning frameworks.

**Program encoding** Programs have rich structural information, which is explicitly represented by their abstract syntax trees (ASTs), e.g., see the AST shown in Figure 3a. Each node in an AST represents an abstract construct in the program source code. We encode programs in such a way that their explicit tree structural information is captured by CNNs easily. To do this, we convert the AST of a program into an adjacency list-like representation as follows. First, we flatten the tree by performing breadth-first traversal. In the second step, each non-terminal node in this flattened tree is replaced by a list, with the first element in the list being the node itself, and the rest of the elements being its direct children, the nodes being ordered from left to right. As terminal nodes do not hold any structure by themselves, we discard them at this step.

Next, we convert this representation into a 2-dimensional matrix for feeding it to a CNN. We do this by padding subtrees with dummy nodes to make them equisized across all programs in our dataset. We also pad the programs with dummy subtrees to make each program have the same number of subtrees. This way, each program is encoded into a 2D matrix of size `max_subtrees` $\times$ `max_nodes`,

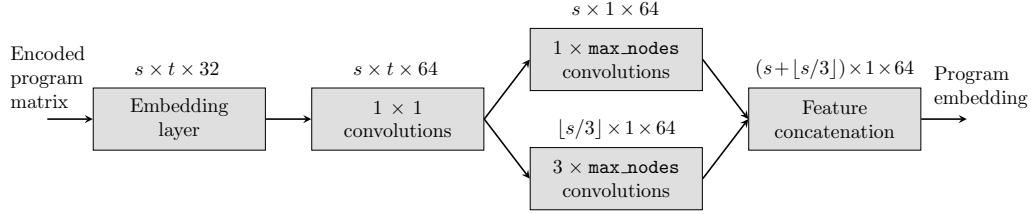

Figure 4: Tree convolution over the encoded program AST input. The input dimensions are $s \times t$, where $s = \texttt{max\_subtrees}$ and $t = \texttt{max\_nodes}$. All three convolutional layers use 'valid' padding.

where $\texttt{max\_subtrees}$ and $\texttt{max\_nodes}$ denote the maximum number of subtrees, and the maximum number of nodes in a depth-1 subtree across all programs in our dataset, respectively. Figure 3b shows the 2D matrix representation for the AST shown in Figure 3a where $0$ indicates padding. In this representation, each row of the encoded matrix corresponds to a depth-1 subtree in the program AST. Moreover, contiguous subsets of rows of an encoded matrix correspond to larger subtrees in the program AST. Note that this encoding ensures that the tree structural information of a program is captured by the spatial neighborhood of elements within a row of its encoded matrix; allowing us to use CNNs with simple convolution filters to extract features from complete subtrees, and not just from any random subset of nodes.

Next, we create a shared vocabulary across all program ASTs in our dataset. The vocabulary retains all AST nodes such as non-terminals, keywords, and literals except for the identifiers (variable and function names) without any modification. Identifiers are included in the vocabulary after normalization. This is done by creating a small set of placeholders, and mapping each distinct identifier in a program to a unique placeholder in our set. The size of the placeholder set is kept large enough to allow this normalization for every program in our dataset. This transformation prevents the identifiers from introducing rarely used tokens in the vocabulary.

**Neural network architecture**   Given a pair of a program and a test ID as input, our learning task is to predict the binary test result i.e., failure or success of the input program on the test corresponding to the given test ID. To do this, we first encode the input program into its 2D matrix representation as discussed above. Each element (node) of the matrix is then replaced by its index in the shared vocabulary which is then embedded into a 32-dimensional dense vector using an embedding layer. The output 3D tensor is then passed through a convolutional neural network to compute a dense representation of the input program, as shown in Figure 4. The first convolutional layer of our network applies filters over a single node at a time. The output is then passed through two independent convolutional layers. The first of these two layers applies filters over one row with a stride of one row. The filters for second one operate on three adjacent rows with a stride of three rows. As discussed earlier, each row of the program encoding matrix represents a depth-1 subtree of the program AST. This makes the last two convolutional layers detect features for one subtree, and three subtrees at a time, respectively. The resulting features from both these layers are then concatenated to get the program embedding. Next, we embed the test ID into a 5-dimensional dense vector using another embedding layer. It is then concatenated with the program embedding and passed through three fully connected non-linear layers to generate the binary result prediction. We call our model tree convolutional neural network (TCNN).

## 3.2 Phase 2: prediction attribution for bug-localization

For a pair of a buggy program and a test ID, such that the program fails the test, our aim is to localize the buggy line(s) in the program that are responsible for this failure. If our trained model predicts the failure for such a pair correctly, then we can query a prediction attribution technique to assign the blame for the prediction to the input program features to find these buggy line(s). As discussed earlier, we use the integrated gradients [25] technique for this purpose.

In order to attribute the prediction of a network to its input, this technique requires a neutral baseline which conveys the complete absence of signal. Sundararajan et al. [25] suggest black images for object recognition networks and all-zero input embedding vectors for text based networks as baselines. However, using an all-zero input embedding vector as a baseline for all the buggy programs does

not work for our task. Instead, we propose to use a correct program similar to the input buggy program (both syntactically and semantically) as a baseline for attribution. This works because the correct program does not have any patterns which cause bugs and hence, conveys the complete absence of the signal required for the network to predict the failure. Furthermore, we are interested in capturing only those changes which introduce bugs in the input program and not the benign changes which do not introduce any bugs. This justifies the use of a similar program as a baseline. Using a very different correct program as a baseline would unnecessarily distribute the output difference to benign changes which would lead to the undesirable outcome of localizing them as bugs.

For a buggy submission by a student, we find the baseline from the set of correct submissions by other students, as follows. First, we compute the embeddings of all the correct programs using our tree CNN. Then we compute the cosine distance of these embeddings from the buggy program embedding. The correct program with the minimum cosine distance is used as the baseline for attribution.

The integrated gradient technique assigns credit values to each element of an embedded AST node, which are then max-pooled to get the credit value for that node. As bug-localization techniques usually localize bugs to the program lines, we further average the credit values for nodes to get the credit values for each line in the program. The nodes corresponding to a line are identified using the parser used to generate the ASTs. We interpret the credit value for a line as the suspiciousness scores for that line to be buggy. Finally, we return a ranked list of program lines, sorted in decreasing order of their suspiciousness scores.

## 4 Experiments

### 4.1 Dataset

For training and evaluation, we use student-written C programs for 29 different programming tasks in an introductory programming course. The problem statements of these tasks are quite diverse, requiring students to implement concepts such as simple integer arithmetic, array and string operations, backtracking, and dynamic programming. Solving these require various language constructs such as scalar and multi-dimensional array variables, conditionals, nested loops, recursion and functions. Our dataset comes with the instructor provided test suite for each programming task. The dataset contains a total of 231 tests across these 29 programming tasks. Note that we work only with the tests written by the instructors of this course and do not write or generate any additional tests. Each program in our dataset contains about 25 lines of code, on average. See Section 1 in the supplement to this paper [12] for more details. A program is tested only against tests from the same programming task it is written for. This is assumed in the discussion henceforth.

We have two classes of programs in our dataset, (1) programs which pass all the tests (henceforth, correct programs), and (2) programs which fail and pass at least one test each (henceforth, buggy programs). We observed that programs which do not pass even a single test to be almost entirely incorrect. Such program do not benefit from bug-localization and hence we discard them. Now for each test, we take a maximum of 700 programs that pass it (including buggy programs that fail on other tests) and a maximum of 700 programs that fail it. Next, we generate subtrees for each of these programs using pycparser [5]. To remove unusually bigger programs, we discard the last one percentile of these programs arranged in the increasing order of their size. Across all the remaining programs, `max_nodes` and `max_subtrees` come out to be 21 and 149, respectively. Pairing these programs with their corresponding test IDs results in a dataset with around 270K examples. We set aside 5% of this dataset for validation, and use the rest for training.

**Evaluation dataset**   Evaluating bug-localization accuracy on a program requires the ground truth in the form of bug locations in that program. As the programs in our dataset come without this ground truth, we try to find that automatically by comparing the buggy programs to their corrected versions. For this, we use Python's `difflib` to find line-wise differences, colloquially referred to as the 'diff', between buggy and correct programs. We do this for every pair of buggy and correct programs that are solutions to the same programming task, and are written by the same student. Note that this is only done to find the ground truth for evaluation. Our technique does not use the corrected version of an incorrect program written by the same student. We include a buggy program in our evaluation set only if we can find a correct program with respect to which its diff is smaller than five lines.

Table 1: Comparison of NBL with three baseline techniques. Top-$k$ denotes the number of buggy lines reported in the decreasing order of their suspiciousness score.

| Technique & Configuration | Evaluation metric | Localization queries | Bug-localization result | | |
|---|---|---|---|---|---|
| | | | Top-10 | Top-5 | Top-1 |
| NBL | $\langle P, t \rangle$ pairs | 4117 | 3134 (76.12%) | 2032 (49.36%) | 561 (13.63%) |
| | Lines | 2071 | 1518 (73.30%) | 1020 (49.25%) | 301 (14.53%) |
| | Programs | 1449 | **1164** (80.33%) | 833 (57.49%) | 294 (20.29%) |
| Tarantula-1 | Programs | 1449 | 964 (66.53%) | 456 (31.47%) | 6 (0.41%) |
| Ochiai-1 | | | 1130 (77.98%) | 796 (54.93%) | 227 (15.67%) |
| Tarantula-* | Programs | 1449 | 1141 (78.74%) | 791 (54.59%) | 311 (21.46%) |
| Ochiai-* | | | 1151 (79.43%) | **835** (57.63%) | **385** (26.57%) |
| Diff-based | Programs | 1449 | 623 (43.00%) | 122 (8.42%) | 0 (0.00%) |

This gives us 2136 buggy programs containing 3022 buggy lines in our evaluation set. Pairing these programs with their corresponding failing test IDs results in 7557 pairs.

In order to identify the buggy lines from the diff, we first categorize each patch appearing in the diff into one of three categories: (1) insertion of correct line(s), (2) deletion of buggy line(s), and (3) replacement of buggy line(s) with correct line(s). Next, we mark all the lines appearing in the deletion and replacement categories as buggy. For the lines in the first category, we mark their preceding line as buggy. For a program with a single buggy line, it is obvious that all the failing tests are caused by that line. However, for the programs with multiple buggy lines, we need to figure out the buggy line(s) corresponding to each failing test. We do this as follows.

For a buggy program and its diff with the correct implementation, first we create all possible partially corrected versions of the buggy program by applying all non-trivial subsets of the diff generated patches. Next, we run partially corrected program versions against the test suite, and for each program, mark the buggy line(s) excluded from the partial fix as potential causes for all the tests that the program fails. Next, we go over these partially fixed programs in the increasing order of the number of buggy lines they still contain. For each program, we mark the buggy lines in that program as a cause for a failing test if all other programs having a strict subset of those buggy lines pass that test. This procedure is similar in spirit to delta debugging approach [31], which uses unit tests to narrow down bug causing lines while removing lines that are not responsible for reproducing the bug.

## 4.2 Training

We implement our technique in Keras [7] using TensorFlow [1] as back-end. We find a suitable configuration of our TCNN model through experimentation Our vocabulary has 1213 tokens after identifier-name normalization. We train our model using the Adam optimizer [17], with a learning rate of 0.0001. We train our model for 50 epochs, which takes about one hour on an Intel(R) Xeon(R) Gold 6126 machine, clocked at 2.60GHz with 64GB of RAM and equipped with an NVIDIA Tesla P100 GPU. Our model achieves the training and validation accuracies of 99.9% and 96%, respectively.

## 4.3 Evaluation

**Classification accuracy of TCNN** In the first phase, we use the trained model to predict the success or failure of each example pair of a buggy program and a test ID, $\langle P, t \rangle$ from the evaluation dataset as input. On these pairs, the classification accuracy of the trained model is only 54.48%. This is much lower than its validation accuracy of 96%. The explanation for such a big difference lies in the way the two datasets are constructed. The pairs in the validation set are chosen randomly from the complete dataset, and therefore, their distribution is similar to the pairs in the training dataset. Also, both training and validation datasets consist of pairs associated with both success and failure classes. On the other hand, recall that the evaluation set contains pairs associated only with the failure class. Furthermore, the buggy programs in these pairs are chosen because we could find their corrected versions with a reasonably small syntactic difference between them. Thus, the relatively lower accuracy of our model on the evaluation set stems from the fact that its distribution is different from that of training and validation sets, and is not actually a limitation of the model. This is also

evident from the fact that the test accuracy increases to about $72\%$ if the evaluation set includes pairs associated with both success and failure classes, instead of just failure class (taken for the same set of programs as in the original evaluation set).

**Bug-localization accuracy** In the second phase, we query the attribution technique for bug-localization of those pairs of programs and tests for which the model prediction in the earlier phase is correct. Towards this, we first find the baseline programs corresponding to the input buggy programs using the method described earlier (in Section 3.2). We use a baseline program only after verifying that the trained network correctly classifies it with respect to the input test ID. If not, we remove it from the set of candidate baseline programs and repeat the procedure until we find a verified baseline program. In our experiments, we could find verified baselines in just one attempt for $97.2\%$ of the pairs, and in two attempts for most of the remaining pairs.

We evaluate the bug-localization performance of NBL on the following three metrics: (1) the number of pairs for which at least one of the lines responsible for the program failing the test is localized, (2) the number of buggy lines localized across all programs, and (3) the number of programs for which at least one buggy line is localized. As shown in Table 1, out of the 1449 programs for which the localization query is made, NBL is able to localize at least one bug for more than $80\%$ of them, when reporting the top-10 suspicious lines per program. NBL also proves to be effective in bug-localization for programs having multiple bugs. Out of 756 such programs in the evaluation set, NBL correctly classified 522 of them as buggy and localized more than one bug for 314 of them, when reporting the top-10 suspicious lines. NBL is also efficient and takes only less than a second to localize bugs in a program, on average.

**Comparison with baselines** Given a program, the actual class (success or failure) for a test can be obtained by executing the program. If the classifier predicts success for a test that actually fails, querying the gradients in that case is unlikely to give a meaningful result. Thus, our approach is to be used when the classification is correct and under this setting, we demonstrate that our approach is competitive with (or even better than several configurations of) human-designed state-of-the-art techniques. In particular, we compare NBL with three baselines including two state-of-the-art program-spectrum based approaches, namely Tarantula [14] and Ochiai [2], and one syntactic difference based approach. The metric used for this comparison is the number of programs for which at least one bug is localized. The other two metrics also yield similar results.

A program-spectrum records which components of a program are covered (and which are not) during an execution. Tarantula and Ochiai compare the program-spectra corresponding to all the failing tests to that of all the passing tests. The only difference between them is that they use different formulae to calculate the suspiciousness scores of program statements. As NBL performs bug-localization with respect to one failing test at a time, we restrict these techniques to use only one failing test at a time for a fair comparison. We use them in two configurations. In the first, they are restricted to just one passing test, chosen randomly; in the second, they use all the passing tests. These configurations are denoted by suffixing '-1' and '-*' to the names of the techniques, respectively.

The syntactic difference based approach is the same as the one used for finding the actual bug locations (ground truth) for the programs in the evaluation set with the exception that the reference implementation for a buggy program submitted by a student is now searched within the set of correct programs submitted by other students. This is done both for this approach, and our technique.

As can be seen in Table 1, NBL outperforms both Tarantula-1 and Ochiai-1 (when they use only one passing test) in top-$k$ results for all values of $k$. However, with the extra benefit of using all passing tests, they both outperform NBL in the top-1 results. Nevertheless, even in this scenario, NBL outperforms both of them in the top-10 results. In the top-5 results, NBL outperforms Tarantula-*, while matching the performance of Ochiai-*. NBL also completely outperforms the syntactic difference based technique with a high margin, demonstrating the effectiveness of learning over a naive syntactic difference based solution.

**Qualitative evaluation** In our analysis, we found that NBL localized almost all kinds of bugs appearing in the evaluation set programs. Some of these include wrong assignments, conditions, for-loops, memory allocations, input reading, output formatting, and missing code among others. This gives credence to our claim that ours is a general semantic bug-localization technique. We provide several concrete examples illustrating our bug-localization results in Section 2 of the supplement [12].

**Faster search for baseline programs through clustering**    As discussed in Section 3.2, we calculate the cosine distance between the embeddings of a given buggy program with all the correct programs solving the same programming task. When the number of correct programs is large, it can be expensive to search through all of them for each buggy program. To mitigate this, we cluster all the programs first using the k-means clustering algorithm on their embeddings. Now for each buggy program, we search for the baseline only within the set of correct programs present in its cluster. Note that both clustering and search are performed on programs from the same programming task. We set the number of clusters to $5$. Our results show that clustering affects the bug-localization accuracy by less than $0.5\%$ in every metric, while reducing the cost of baseline search by a factor of $5$.

## 5    Related work

Program repair techniques are extensively used for feedback generation for logical errors in student programs. AutoGrader [24] takes as input a buggy student program, along with a reference solution and a set of potential corrections in the form of expression rewrite rules and searches for a set of minimal corrections using program synthesis. Refazer [22] learns programs transformations from example code edits made by students using a hand designed domain specific language, and then uses these transformations to repair buggy student submissions. Unlike these approaches, our approach is completely automatic and requires no inputs from the instructor. Most program repair techniques first use an off-the-shelf bug-localization technique to get a list of potential buggy statements. On these statements, the actual repair is performed. We believe that our technique can also be fruitfully integrated into such program repair techniques.

Another common approach to feedback generation is program clustering where student submissions having similar features are grouped together in clusters. The clusters are typically used in the following two ways: (1) the feedback is generated manually for a representative program in each cluster and then customized to other members of the cluster automatically [19, 20, 8], and (2) for a buggy program, a correct program is selected from the same cluster as a reference implementation, which is then compared to the buggy program to generate a repair hint [15, 10, 27, 23]. The clusters are created either using heuristics based on program analysis techniques [8, 15, 10, 27, 23] or using program execution on a set of inputs [19, 20]. Unlike these approaches, we cluster programs using k-means clustering algorithm on program embeddings learned on program ASTs, which does not require any heuristics and therefore, is programming language agnostic.

## 6    Conclusions and future work

We present the first deep learning based general technique for semantic bug-localization in student programs with respect to failing tests. At the heart of our technique is a novel tree convolution neural network which is trained to predict whether or not a program passes a given test. Once trained, we use a state-of-the-art neural prediction attribution technique to find out which lines of the programs make the network predict the failures to localize the bugs. We compared our technique with three baseline including one static and two dynamic bug-localization techniques. Our experiments demonstrate that our technique outperforms all three baselines in most of the cases.

We evaluate our technique only on student programs. It will be an interesting future work to use it for arbitrary programs in the context of regression testing [30], i.e., to localize bugs in a program w.r.t. the failing tests which were passing with the earlier version(s) of that program. NBL is programming language agnostic and has been evaluated on C programs. In future, we will experiment with other programming languages as well. We also plan to extend this work to achieve neural program repair. While our bug-localization technique required only a discriminative network, a neural program repair technique would require a generative model to predict the patches for fixing bugs. It will be interesting to see if our tree CNN can be adapted to do generative modeling of patches as well.

**Acknowledgments**

We gratefully acknowledge Sonata Software Ltd. for partially funding this work. We thank Prof. Amey Karkare and his research group from IIT Kanpur for making the dataset available. We also thank the anonymous reviewers for their helpful feedback on the earlier versions of the paper.

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
