[Supplementary Material]

# Neural Attribution for Semantic Bug-Localization in Student Programs

## Supplementary Material

**Rahul Gupta**[1]    **Aditya Kanade**[1,2]    **Shirish Shevade**[1]
[1]Department of Computer Science and Automation,
Indian Institute of Science, Bangalore, KA 560012, India
[2]Google Brain, CA, USA
{rahulg, kanade, shirish}@iisc.ac.in

## 1   Dataset

Table 1: Dataset statistics.

| Avg. programs per task | | Avg. tests per task | Avg. submissions per student | Avg. lines per submission |
|---|---|---|---|---|
| Correct | Buggy | | | |
| 350 | 1007 | 8 | 18 | 25 |

We train and evaluate NeuralBugLocator on C programs written by students for 29 different programming tasks in an introductory programming course. The dataset comes with 231 instructor written tests for these tasks. On an average, each programming task contains about 350 correct and 1007 buggy programs, and 8 instructor written tests. The number of average submissions made per student is about 18. On an average, each submission has about 25 lines of code. Table 1 shows the dataset statistics. Below we list the problem statements for some of the programming tasks.

---

You want to create an intelligent machine which can perform linear algebra for you. In linear algebra, we often encounter identity matrices. Therefore, teaching computers to recognize whether a matrix is identity or not is one of the tasks that you must perform in your quest to build such a machine. In this problem, you'll write a program to check whether a given matrix is identity or not.

In the first line, you'll be given $n$, which will be the number of rows and number of columns in identity matrix. In the next $n$ lines, you'll be given entries of the matrix with each row in a new line. If the matrix is identity, then print "GIVEN $n \times n$ matrix is an IDENTITY MATRIX". Otherwise, print "GIVEN $n \times n$ matrix is NOT an IDENTITY MATRIX". Here, $n$ is the dimension of the matrix.

Note: You are not allowed to use arrays to store the input.

---

Factors of a numbers are often required to know about the characteristics of a number.

---

In this problem, you'll print all prime factors of a given integer. (Prime numbers are the numbers which have exactly two factors i.e. 1 and itself). You have to print all prime factors of a number in a new line in descending order. If a number is itself prime, print $-1$.

Write a program to implement a rotation cipher as defined:

The program first reads three integers $k1$, $k2$ and $k3$ separated by white spaces. It then reads a characters from the NEXT line. Change the character according to the following rules:

    (a) if it is a lower case character, it is rotated by $k1$ positions. For example, if $k1$ is 3 then 'a' becomes 'd', 'b' becomes 'e', ..., 'x' becomes 'a', 'y' become 'b', 'z' becomes 'c'.

    (b) if it is an upper case character, it is rotated by $k2$ positions. For example, if $k2$ is $-3$ then 'A' becomes 'X', 'B' becomes 'Y', ..., 'X' becomes 'U', 'Y' become 'V', 'Z' becomes 'W'.

    (c) if it is a digit, it is rotated by $k3$ positions. For example, if $k3$ is 4, then 3 becomes 7, 6 becomes 0, ..., 0 becomes 4, 5 become 9 and so on.

    (d) any other character remains the same.

The output is a single character obtained after above change.

Given two integer arrays (let's say A1 and A2), check if A2 is a contiguous subarray of A1. A2 is a contiguous subarray if all elements of A2 are also present in A1 in the same order and continuously.

For example, $[12, 42, 67]$ is a contiguous subarray of $[1, 62, 12, 42, 67, 96]$ Whereas, $[1, 23, 21]$ and $[12, 42, 96]$ are not contiguous subarrays of $[1, 62, 12, 42, 67, 96]$

Input: The first line contains the size $N1$ of first array. Next line contains $N1$ space separated integers giving the contents of first array. Next line contains the size $N2$ of second array. Next line contains $N2$ space separated integers giving the contents of second array.

Output: Either YES or NO (followed by a new line).

Variable Constraints: The array sizes are smaller than 20. Each array entry is an integer which fits an int data type.

You are given two integers $n1$ and $n2$ followed by two space separated strings 'str1' and 'str2' of length $n1$ and $n2$ respectively, each consisting of lowercase characters. The length of each of the strings is not more than 500.

Output the length of the initial segment of str1 which consists entirely of characters in str2.

You are given an array of n numbers. You have to find out whether the array is a SuperArray or not. An array is a SuperArray if it satisfies the following constraints.

Every element $A[i]$ of the array should occur $A[i]$ times. For example if the array contains 2, then there should be exactly two occurrences of the number 2 in the array.

Find out whether or not a path exists through a given maze. The maze is a 2D matrix where '.' denotes path and 'X' denotes wall. It starts at $(0, 0)$ and end at the bottom-right (both of which will always be '.')

Input: Space separated integers $m$, $n$ denoting size of matrix. Next $m$ lines contain a string of $n$ characters (composed of '.' and 'X')

Input Constraints: $1 \leq m, n \leq 15$

Output: YES if path exists, NO otherwise

In this exercise, you need to implement GCD. However, the challenge is that you are not allowed to return any values. So, the modified GCD function takes two pointers as follows: `void gcd(int *a, int *b)`. It modifies the values such that when the function returns, a will contain the final answer. You need to use the function signature from the initial template.

Write a program to find $k^{th}$ largest element of an array.

Full points will only be awarded if your solution is based on repeated applications of the Partition function (which was introduced for QuickSort). You do not have to sort the whole array, as this will fetch you half of the total points.

Any other solution e.g., solutions based on sorting, etc. will at most fetch half of total points.

Please see the provided template for hints.

Input will have two lines — The first line will have an integer n denoting number of elements of the array and k; next line will contain n space separated integers denoting the elements of the array.

Output: you have to return the $k^{th}$ largest element of the array

The professor of PHY101A has decided to catch all cheating cases. Since you have already done that course, you decide to help him in this task by automating his work.

You are going to calculate the 'proximity' between any 2 documents by counting the longest common substring in the 2 documents. For example, - If one of the document is 'ABA' and the other document is 'BAB', the proximity is 2 since the longest common substring is 'AB' (or 'BA'). - If one of the document is 'doc1' and the other document is 'doc2', the proximity is 3 since the longest common substring is 'doc'.

Input: Two integers ($n1$ and $n2$) denoting the length of first and second document. Content of first document ($n1$ characters) Content of second document ($n2$ characters)

Output: A single integer, the proximity between the documents

## 2 Concrete Examples Illustrating NeuralBugLocator Results

**Incorrect Input Reading and Output Formatting**

```c
#include <stdio.h>
int main(){
    int n,i;
    char c;
    scanf("%d",&n); \\ suspiciousness score: 0.0007697232
    for(i=0;i<n;i++) {
        scanf("%c",&c);
        if (c=='a'|| c=='e' || c=='i' || c=='o'|| c=='u') {
            printf("Special");
            printf("\n%d",i); \\ suspiciousness score: 0.00045288168
            break; } }
    if(i==n)
    printf("Normal");
    return 0; }
```

This program is supposed to print 'Special' if the given input string contains a vowel, otherwise 'Normal'. The input format is an integer 'n' and a string 's' of length n, separated by a newline

character. However, the `scanf` function in line 7 reads the newline character following 'n' as the first character of the string. Therefore, if the input string is a vowel, that will not be read and the program will print the wrong output 'Normal'. One way to fix it is to append the newline character after the "%d" format specifier in the `scanf` function call of line 4. Also, there is an additional print statement at line 10 which prints spuriously, causing an output mismatch. NeuralBugLocator ranks these two as its third and fourth most suspicious buggy lines, respectively.

## Wrong Condition

```c
1  #include <stdio.h>
2  int main() {
3      int n,i;
4      char a[100];
5      char b;
6      int flag=0;
7      scanf ("%d",&n);
8      for (i=0;i<n;i=i+1) {
9          scanf ("%c",&b);
10         if((b=='a')||(b='e')||(b='i')||(b=='o')||(b=='u')) \\ suspiciousness
                   score: 0.0015987115
11         flag=1; }
12     if(flag==1) {
13         printf("Special"); }
14     else {
15         printf ("Normal"); }
16     return 0; }
```

The program shown above solves the same problem as the program last discussed. It twice uses the assignment operator instead of the comparison operator in line 10 which causes the bug. NeuralBugLocator localizes it in its top prediction.

---

```c
1  #include<stdio.h>
2  int main() {
3      int a,b,c;
4      scanf("%d%d%d",&a,&b,&c);
5      if (a+b>c) {
6          if (a*a+b*b==c*c){printf("RIGHT");}
7          else if(a*a+b*b<c*c||a*a>b*b+c*c||a*a+c*c<b*b){ \\ suspiciousness score
                   : 0.0004023624
8        printf("OBTUSE" );}
9          else if(a*a+b*b>c*c||a*a<b*b+c*c||a*a+c*c>b*b){ \\ suspiciousness score
                   : 0.00045172646
10       printf("ACUTE");} }
11     else if(a+b==c) {printf("INVALID");}
12     return 0; }
```

The above program is supposed to check and print if a triangle is invalid, acute, right or obtuse, given the length of its three sides. However, the conditions used in lines 7 and 9 are buggy. To fix the program, they should be replaced by lines (1) `else if(a+b>c && a+c>b && b+c>a) {` and (2) `else {`, respectively. NeuralBugLocator ranks them as its third and fourth most suspicious buggy lines, respectively.

**Insufficient memory allocation**

```c
#include <stdio.h>
int in(int k,int n,int l[100]){
    int i;
    for(i=0;i<n;i++){
        if(l[i]==k){
            return 1; } }
    return 0; }
int main(){
    int n;
    int ip[100];
    int u[100]; \\ suspiciousness score: 0.0012746735
    scanf("%d",&n);
    int i;
    for(i=0;i<n;i++){
        scanf("%d",&ip[i]); }
    int k=1,count=0;
    i=0;
    while(!in(k,n,u)){
        u[i]=k;
        k=ip[k-1];
        i+=1;
        count+=1; }
    printf("%d ",count);
    for(i=0;i>=0;i++){
        if(u[i]==k){
            printf("%d",count-i);
            break; } }
    return 0; }
```

The program shown declares an array of fixed-size in line 11 which fail on tests containing larger inputs. NeuralBugLocator localizes the buggy statement it in its top prediction. Note that the other fixed-sized array declared in line 10 is not considered buggy as it does not cause any available test to fail.

**Type Narrowing**

```c
#include <stdio.h>
int main(){
    int x1,y1,x2,y2;
    float slope;
    scanf("%d%d%d%d",&x1,&y1,&x2,&y2);
    if(x1==x2) {
        printf("inf");
        return 0; }
    else {
        slope==(y2-y1)/(x2-x1); \\ suspiciousness score: 0.0028934027
        printf("%.2f\n", slope); }
    return 0; }
```

This program calculates the slope of a line specified by two points whose coordinates are given as four integers $(x_1, y_1)$ and $(x_2, y_2)$. When calculating slope in line 10, the division of integers returns integer value and not a floating point value. This is known as narrowing of types and can be fixed with type-casting any of the variable or expression in the RHS of the assignment as float before the division operation. The buggy line also mistakenly uses a comparison operator instead of the assignment operator. NeuralBugLocator localizes the buggy line in its top prediction.

**Wrong Assignment**

```c
#include <stdio.h>
#include<string.h>
int main() {
    int i,j,c;
    char str1[10],str2[10];
    scanf("%s %s",str1,str2);
     c=strlen(str2);
    for(i=0;str1[i]!='\0';i++) {
       str1[i]=str1[i]+str2[i%c]-'a'+1; } \\ suspiciousness score: 0.0011707128
  printf("%s",str1);
    return 0; }
```

The program shown above is supposed to shift a string by another pattern string (both given as input) and print the result. However, the RHS expression of the assignment at line 9 is buggy. The correct RHS expression is: (str1[i]+str2[i%c]-'a'-'a'+1)%26+'a';. NeuralBugLocator localizes the buggy line in its top prediction.

---

```c
#include <stdio.h>
int main(){
    int a,b,i,n,m;
    scanf("%d%d%d",&a,&b,&m);
    n=1;
    for (i=1;i<=b;i=i+1)
    n=n*a; \\ suspiciousness score: 0.007239882
    printf("%d",n%m);
  return 0; }
```

The program shown above implements $a^b \bmod m$. However, the RHS expression in line 7 does not implement this logic correctly. The fix for this line is: n=(n*a)%m;. NeuralBugLocator localizes the buggy line in its top prediction.

**Missing Code**

```c
#include <stdio.h>
int main(){
    int n,max=0,sum,i,j=0;
    scanf("%d/n",&n);
    char s[n],ch;
    ch=getchar();
    for(i=0;i<n;i++)
    {ch=getchar();
    s[i]=ch;}
    for(i=0;i<n;i++)
    { sum=0; \\ suspiciousness score: 0.0013130781
        while(s[i]==s[i+j])
        {sum++;
        j++; }
        if(max<=sum)
        max=sum; }
    printf("%d",max);
    return 0; }
```

This program is written for finding the longest contiguous streak of a character in a given string. To correctly implement this, the programmer needs to insert j=0; at line 11. NeuralBugLocator localizes this bug in its top prediction.

**Wrong `for` Loop**

```c
#include <stdio.h>
int rot(int [],int,int);
int main() {
    int n,d,i;
    scanf("%d\n",&n);
    int arr[n];
    for(i=0;i<n;i++) {
        scanf("%d ",&arr[i]); }
    scanf("\n%d",&d);
    rot(arr,n,d);
    return 0; }

int rot(int arr[],int n,int d) {
    int j,k;
    for(j=d+1;j<n;j++) { \\ suspiciousness score: 0.0006181474
        printf("%d ",arr[j]); }
    for(k=0;k<=d;k++) { \\ suspiciousness score: 0.0006690205
        printf("%d ",arr[k]); }
    return 0; }
```

This program is supposed to right shift a given array of 'n' numbers by a given number 'd'. To correctly implement this, the programmer needs to change the two `for` loops at lines 15 and 17 to `for(j=n-d;j<n;j++)` and `for(k=0;k<n-d;k++)`, respectively. NeuralBugLocator ranks these two lines as its second and third most suspicious buggy lines, respectively.