[Reviews · NeurIPS 2019]

Reviewer 1



This work targets the localization of bugs in student programs. This is accomplished by training an CNN to, given the input source code, predict whether that program passes given test cases. For failing tests, the Integrated Gradients technique is then used to try and attribute the prediction back to specific lines of code with the intention being that these lines are likely to be responsible (ie, buggy). One key trick here is an AST-based featurization that "spatializes" the data such that adjacent rows correspond to subtree structure (Section 3.1.1). Encoding the tree-based structure of the target code into a spatial-like matrix in this way has the advantage of allowing the use of an "off the shelf" CNN implementation, with the accompanying engineering advantages. The idea of using attribution or interpretability in this way is potentially novel and significant. However the submission could be improved with some clarifications, more detail, and a more explicit characterization of the dependencies on the particular properties of the student program problem domain data. MORE DETAILED COMMENTS Figure 3: this small worked example was essential for understanding Section 3.1.1. L159: if I'm understanding correctly, there are unique tokens for each of the node contents from Figure 3 ("IdentifierType:int", "BinaryOp:%", and so on), but then there is some special "normalization" done for variable names like "Decl:even" or "ID:num"? The mechanics of this normalization step were not clear to me. The mechanics of the initial embedding layer are not very clear from the paper either, but I'm guessing it is just using the Keras Embedding layer, which is ultimately jointly trained along with the full classifier network? So, the input to Supplement Figure 1 ends up being (24 x max_subtrees x max_nodes)?? Supplement Figure 1: this contains insufficient detail to easily understand the model. It would help to track the current dimensionality at each subcomponent, or use some other representation or notation to capture more granular details of what's going on here. Section 3.1.2: how was this architecture determined? Was it derived from some mixture of general intuition plus trial-and-error, based on other related works, or some other method? Section 3.1.2: wouldn't stride 3 cause us to "miss" subtrees such as the one rooted at Node 3 in Figure 3? That is, we'd have a convolution for the tree rooted at Node 1, then stride all the way to the subtree rooted at Node 5. L45: the summary of bug localization techniques as static vs dynamic is a helpful framing, and it was good to see both families of approaches represented in the experimental baselines.] L120: "Note that..." - this is a key aspect of the approach and useful to emphasize here. L194: the requirement for a "nearby" correct program might not be easy to meet in other application domains, is there any idea how robust the NeuralBugLocator would be to using less similar correct programs for the attribution? Section 4.3: it makes sense that the classification accuracy would be lower for the "nearly correct" programs in the evaluation set. It seems a little strange to restrict the evaluation against baselines to only cases which were classified correctly though, throwing away nearly half of the evaluation set? In a "real" system, these would simply be misses or failures for NeuralBugLocator. Also, the sense in which Tarantula and Ochiai use the passing test data is not very clear - maybe some more detail about how they are using the input data would clarify this. Table 1: it is confusing to mix the "test id" and "Lines" granularities with the "Programs" results in this table. UPDATE: I have read the author response and found the clarifications and additional details helpful. I believe the submission would be enhanced by the revisions proposed by the authors in the response.

Reviewer 2



This paper addresses finding bugs in short pieces of code. The dataset used in the paper is created from student's submissions/instructor's tests to a introductory programming class. The technique consists in two parts: (1) a component that predicts which programs are buggy (fail a test) without running the program by training a convolutional network that operates on flattened code ASTs. I found the encoding of the programs/ASTs to matrices and the use of CNN quite interesting. I'm guessing it might be faster to train such networks than the RNNs that are usually employed when representing/embedding code. The application of prediction attribution to bug localization is also interesting and most likely novel. The paper is generally well written and straightforward until the evaluation section. Please see comments below. ================ UPDATE: Thank you for your feedback. I encourage the authors to polish the experimental section which is hard to follow. The explanations in the feedback helped me better understand the work, thank you. In addition, I would state - even if informally - the experience with running RNNs on this problem. Thank you for confirming my intuition that the RNNs are super-slow to train for this type of problems. I think it's important to communicate this type of results. Currently we have few/no venues to communicate "negative" results, and, IMO "negative" results are as important (if not more important in some cases...) as "positive" results. (I'm using quotes because everything is relative). I'm increasing my overall score from 6 to 7 in the light of the clarifications during author feedback.

Reviewer 3



First, I should say that the paper is very novel in comparison to many of the works that apply deep learning for programming tasks. From methodological perspective this may be the best paper in the area for 2019. The idea to train a classifier on whether a test will succeed and then to use feature attribution techniques for localizing the errors in the program is new, original and quite interesting. I also expect that despite the limited use case shown in the current paper, the idea is much larger and should be tried in a broader set of scenarios. The paper is also well written, well motivated and should be of significant interest to most researchers working on learning for code. Furthermore, it provides good details both on the high-level idea, as well as on the detailed of the neural architecture. In terms of evaluation, while results are not completely conclusive that the technique is superior to the state-of-the-art in all scenarios, it shows good promise. One should keep in mind, however, that the comparison is not to naive neural baselines, but to complex specialized systems to perform similar tasks.

[Author Response · NeurIPS 2019]

We thank the reviewers for detailed comments and helpful suggestions. We will incorporate them to improve the paper.

**Reviewer 1 — To normalize variable (and function) names**, we create a vocabulary of $N$ new identifiers. For each program, we derive a random map from variable names to the new identifiers, and rename all occurrences of a variable with its corresponding identifier. This ensures that the program semantics does not change. We ensure that the size $N$ of our vocabulary is larger than the number of variables in any program in the dataset. This method of reducing vocabulary sizes has been used before in the literature (e.g., [11]).

**Network architecture:** The reviewer's summary of the embedding layer is correct (L170-L171). We will add more details in the figure as suggested by the reviewer. The first model we tried was more complex and had two more convolutional filters overlapping 2 and 4 rows of the encoded matrix with the strides of 2 and 4, respectively. Through experimentation, we found that removing those filters did not affect the performance of the network but helped in increasing training efficiency. We also tried RNNs (please see response to reviewer#2).

**Stride** size 3 is used for a convolutional filter spanning 3 rows. So subtree rooted at Node 3 will be analyzed as part of the first 3 rows (Fig 3b). Separately a filter spanning 1 row at a time with stride of 1 (L175-L176) will also cover it.

**Comparison with baselines:** Given a program, the actual class (success or failure) for a test can be obtained by executing the program. If the classifier predicts success for a test that actually fails, querying the gradients in that case is unlikely to give a meaningful result. Thus, our approach is to be used when the classification is correct and under this setting, we demonstrate that our approach is competitive with (or even better than several configurations of) human-designed SOTA approaches. We will elaborate on this in the paper.

**Use of passing tests in Tarantula and Ochiai:** Tarantula and Ochiai use coverage metrics over lines (called program spectra), obtained by executing passing and failing tests, and calculate the suspiciousness score for each line based on some empirical formulae. The different configurations in Table 1 indicate how many passing tests were used in the comparison. We will explain this more.

**Generalizability to other settings:** We exploit similarity between code along with prediction attribution for semantic bug localization and demonstrate it for student code. In the industrial setting also, code similarity abounds due to code reuse and cloning. Large scale studies (see "On the naturalness of software", ICSE'12) have demonstrated that code tends to be quite repetitive (similar to natural language utterances). The practice of version controlling results in similar but evolving copies of code, which are typically subjected to regression testing (L341). Though more experimentation will be required, we expect our approach to be useful in these settings.

**Reviewer 2 — Comparison with RNNs:** We experimented with an attention-based LSTM network for failure prediction. Training it took more than two days and the performance of prediction attribution was not as good. In comparison, the proposed tree CNN took only one hour to train (L252) and enabled better attribution.

**Use of test IDs vs the complete tests:** Embedding a unit test along with the program is a great suggestion. This can improve prediction accuracy and attribution, particularly when test code implements some protocols to set up the input objects (such as files). However, in our current setup involving student code, the tests consist of raw inputs and outputs, and lack useful structure. We therefore use only test IDs.

**Evaluation section and significance of results:** We will reword the evaluation section to make it more clear. Our results show that our technique is competitive to the SOTA dynamic bug-localization techniques which require program instrumentation and collecting program-spectra through multiple executions. We also show that it completely outperforms a naive static approach that uses syntactic difference between a buggy program and its reference implementation for bug-localization.

The percentages shown in the Table 1 correspond to **recall**. **Precision** can be calculated by dividing the number of lines localized by the number of predictions made ($=$ number of programs multiplied by $k$, where $k$ is the number of suspicious lines reported). Precision values come out to be $0.1$, $0.14$, and $0.21$ when $k$ is set to $10$, $5$, and $1$ respectively.

**Scaling to larger programs:** We envision the use of our technique at the level of unit tests where methods are tested individually. While method bodies can be large at times, typically they are (encouraged to be) short. Nevertheless, owing to the fast training possible for tree CNN (L252), we are positive about scaling our technique to larger programs.

**Reviewer 3 — Classification w.r.t. one test:** As the reviewer points out, our classifier analyzes one test at a time. It is an interesting future direction to do localization using entire test suites instead. The dataset-level attribution methods (e.g., based on clustering), called global attribution methods, will be useful in this context.

**Runtimes for bug localization:** It takes us $4.69$ seconds for calculating the embeddings for $8086$ correct programs across all the programming tasks ($0.5$ ms per program). For finding attribution baseline and then performing bug-localization through attribution, it takes about $0.67$ seconds per program. We will add these to the paper.

[Meta-Review · NeurIPS 2019]

The paper received excellent reviews and strong acceptance recommendations. However interesting the targeted application is, I personally found that the technical content was low and the level of the experimental section close to the average of the standards of Machine Learning applications. Therefore I recommend acceptance as a poster.